# Human IL12p80 Promotes Murine Oligodendrocyte Differentiation to Repair Nerve Injury

**DOI:** 10.3390/ijms23137002

**Published:** 2022-06-23

**Authors:** Yu-Fen Chung, Jong-Hang Chen, Ching-Wen Li, Hui-Yu Hsu, Ya-Ping Chen, Chiao-Chan Wang, Ing-Ming Chiu

**Affiliations:** 1Institute of Cellular and System Medicine, National Health Research Institutes, Miaoli 350, Taiwan; yufen@nhri.edu.tw (Y.-F.C.); j.h.chen588@gmail.com (J.-H.C.); chingwenli@psi.com.tw (C.-W.L.); kakacoco0218@gmail.com (H.-Y.H.); 090406@nhri.edu.tw (Y.-P.C.); 060118@nhri.edu.tw (C.-C.W.); 2Biotechnology Center, National Chung Hsing University, Taichung 402, Taiwan; 3Graduate Institute of Biomedical Sciences, Neuroscience and Brain Disease Center, China Medical University, Taichung 404, Taiwan; 4Department of Internal Medicine, The Ohio State University, Columbus, OH 43210, USA

**Keywords:** human IL12p80, neural stem cells, sciatic nerve injury, nerve regeneration, nerve conduits

## Abstract

Nerve injury of the central nervous system and the peripheral nervous system still poses a major challenge in modern clinics. Understanding the roles of neurotrophic factors and their molecular mechanisms on neuro-regeneration will not only benefit patients with neural damage but could potentially treat neurodegenerative disorders, such as amyotrophic lateral sclerosis. In this study, we showed that human IL12 p40-p40 homodimer (hIL12p80) within PLA and PLGA conduits improved sciatic nerve regeneration in mice. As such, the group of conduits with NSCs and hIL12p80 (CNI) showed the best recovery among the groups in the sciatic functional index (SFI), compound muscle action potential (CMAP), and Rotarod performance analyses. In addition, the CNI group had a faster recovery and outperformed the other groups in SFI and Rotarod performance tests beginning in the fourth week post-surgery. Immunohistochemistry showed that the CNI group increased the diameter of the newly regenerated nerve by two-fold (*p* < 0.01). In vitro studies showed that hIL12p80 stimulated differentiation of mouse NSCs to oligodendrocyte lineages through phosphorylation of Stat3 at Y705 and S727. Furthermore, implantation using PLGA conduits (C2.0 and C2.1) showed better recovery in the Rotarod test and CMAP than using PLA conduits in FVB mice. In B6 mice, the group with C2.1 + NSCs + hIL12p80 (C2.1NI) not only promoted sciatic functional recovery but also reduced the rate of experimental autotomy. These results suggested that hIL12p80, combined with NSCs, enhanced the functional recovery and accelerated the regeneration of damaged nerves in the sciatic nerve injury mice. Our findings could further shed light on IL12′s application not only in damaged nerves but also in rectifying the oligodendrocytes’ defects in neurodegenerative diseases, such as amyotrophic lateral sclerosis and multiple sclerosis.

## 1. Introduction

Peripheral nerve injury causes nerve fiber degeneration and degradation of surrounding tissues, leading to reduced motor and sensory activities. The regeneration of injured peripheral nerves is a multiplex process, with Schwann cells playing an importing role [1]. After the injury, resident Schwann cells transdifferentiate into proliferation-repairing cells and secrete neurotrophins and rescue macrophages to clear debris [2,3,4]. In clinical application, nerve conduits provide mechanical support and direct axonal sprouting between the injured nerve stumps [5]. Conduits have been shown to retain neurotrophic factors secreted from or recruited by the damaged cells and prevent ingrowth of fibrous tissue at the injured site [5]. Recent studies have revealed that the implantation of neural stem cells (NSCs) promotes the regeneration of the injured peripheral nerve. The implanted NSCs differentiate into Schwann cells, secrete neurotrophic factors, enrich the microenvironment from the milieu, and assist in myelination [1,6,7,8].

Interleukin-12 (IL12) is a multifunctional cytokine that is naturally produced by macrophages and dendritic cells [9]. It was identified as a heterodimeric (IL12p70, composed of p35 and p40 subunits) protein, which was joined by disulfide bonds. In rare situations, IL12p80 (composed of p40 and p40 monomers) can also be found. The dimeric IL12p70 and IL12p80 and the monomeric IL12p40 are known as secretory proteins. IL12p70 and IL12p80 are crucial in inflammatory and anti-inflammatory responses, respectively [10,11,12]. IL12p80 or IL12p40 is bound to the IL12 receptor β1 (IL12Rβ1), which resulted in antagonizing IL12p70 binding in mice and humans [13,14,15,16]. During the Wallerian degeneration, IL12 was secreted by macrophages for a process of axonal regeneration. Expression of IL12p40 mRNA was detected between days 7 and 14 of Wallerian degeneration [17,18,19]. In a sciatic nerve injury FVB mouse model, we identified IL12p80 protein in the cell extracts of implanted poly(L-lactic acid) (PLA) conduit combined with NSCs by protein antibody array and Western blotting [20]. We also showed that mouse IL12p80 (mIL12p80) triggers mouse KT98/F1B-GFP^+^ NSCs to differentiate into Schwann cells, enhancing the functional recovery and enlarging the diameter of regenerated nerves in FVB mice [20].

To develop human IL12p80 (hIL12p80) as a biomedicine for nerve regeneration, the alignment of amino acid sequences of human and mouse IL12p40 showed 74% conserved sequence and 65% identical sequence. This high degree of conservation between the human and mouse IL12p40 prompted us to test if human IL12p40 could enhance the differentiation of mouse NSCs and the concomitant repair of sciatic nerve injury in mice. Poly(L-lactic-co-glycolic acid) (PLGA) has been used as a material for nerve conduit due to its ease of fabrication, low inflammatory response [21], and approval by the FDA. We have previously shown that the PLGA conduits with micro/nanohybrid patterns and microfibrous structures can retain their shapes and demonstrate a high capability for guiding nerve cells and promoting cell migration in vitro [22]. Therefore, we devised a combination of NSCs, hIL12p80, and the micro-/nanohybrid with a microfibrous structured PLGA conduit to determine the regeneration effect in repairing sciatic nerve injury in mice. Here, we verified that administration of hIL12p80 was beneficial in the repairing of sciatic nerve injury in a FVB mouse model. The micro/nanohybrid-structured PLGA conduits, with or without microfiber, were seeded with NSCs and hIL12p80, and demonstrated their benefit in the functional recovery of sciatic nerve injury in the FVB mice. Furthermore, we showed that implantation of hIL12p80 and NSCs with the micro/nanohybrid-structured PLGA nerve conduit was beneficial in the enlargement of the regenerated sciatic nerve and accompanied with functional recovery of the sciatic nerve injury in the B6 mouse model. Our results could pave the way for developing hIL12p80 as a biologic in repairing sciatic nerve injury in human.

## 2. Results

### 2.1. Implantation of Human IL12p80 and NSCs within PLA Conduits Improved Functional Recovery and Nerve Conduction of Sciatic Nerve Injury FVB Mice

In this study, a schematic diagram of the sciatic nerve injury regeneration FVB mouse model (Figure 1A) and all of the different treatments for improving sciatic nerve injury were presented (Figure 1B) and PLA conduits were applied for the implantation surgery. Compared to the pre-surgery, the Sham group presented no significance in SFI scores from the first to the eighth week after surgery. During the first two weeks after surgery, the SFI scores between the Neg., C, CI, CN, and CNI groups showed no significant difference, but were very different from the Sham group (Figure 1C). However, beginning from the fourth week after surgery, the CN and CNI groups exhibited significantly higher scores in SFI than the Neg. group (Figure 1C). On the sixth and eighth weeks after surgery, the C, CI, CN, and CNI groups showed significantly higher SFI scores than the Neg. group (Figure 1C). Notably, the CNI group showed greater statistical significance than the C and CI groups on the eighth week after surgery (Figure 1C). On the fourth week after surgery, the CNI group showed a significant difference compared with the Neg., C, and CN groups, but no significance to the CI group. The CNI group performed the longest Duration on Rotarod among the Neg., C, CI, CN, and CNI groups (Figure 1D). On the eighth week after surgery, the CNI group had the best Duration on Rotarod test compared to the other groups (Figure 1E). Overall, except for the Neg. group, all four treatment groups improved from the fourth week to the eighth week. In particular, the CNI group outperformed even the Sham group by 67%. In the Recovery of CMAP analyses, the CNI group performed better than any other groups in the order of CNI > CN > CI > C > Neg. (Figure 1F). These results suggested that implantation of human IL12p80 and NSCs within PLA conduits (CNI group) performed better than all other groups in SFI scores, Rotarod analyses, and the Recovery of CMAP in the sciatic nerve regeneration FVB mouse model.

### 2.2. Implantation of hIL12p80 and NSCs with PLA Conduits Promoted Regeneration of Injured Nerves in FVB Mice

In the H&E staining, the CN and CNI groups showed significant enhancement in the regeneration of the sciatic nerve compared with the C group (Figure 2A–D). The level of nerve regeneration was further verified by immunohistochemistry (IHC) using anti-NF200 (a neuron marker) and anti-PZ0 (a myelin Schwann cell marker) antibodies. The results demonstrated that the coupling of myelinating Schwann cells (PZ0- positive cells) with the nerve fiber (NF200-positive cells) of the regenerated nerves in the conduits (dotted square regions in the H&E staining) occurred in the CI (Figure 2F), CN (Figure 2G), and CNI (Figure 2H) groups, but few were in the C group (Figure 2E) in the order of CNI > CN > CI > C. To highlight the immunostaining of PZ0 in myelinating Schwann cells, we also showed the composite staining of PZ0 and DAPI, without the immunostaining of NF200. The results showed that PZ0 staining in the CNI group (Figure 2L) was much more intense than in the C (Figure 2I), CI (Figure 2J), and CN (Figure 2K) groups. The NF200 signals are also populated throughout the newly regenerated axons, particularly in the CNI group. It is interesting that the newly regenerated neurons in the inner regions are associated with myelinating Schwann cells, while the neurons in the peripheral regions exhibited much fewer PZ0 signals (Figure 2H,L).

Furthermore, we measured the mean diameters of regenerated nerves by reconstructing the consecutive tissue sections stained by H&E in proximal (1.0 mm from the proximal site of a conduit), distal (3.0 mm from the proximal site of a conduit), and medial (the center between the proximal and distal sites) sites of implanted nerve conduits (Figure 2M). The schematic diagrams of the regenerated nerves with the four groups are illustrated in Figure 2N. In the medial regions of the four groups, the mean diameter of regenerated nerves in the CNI group was significantly higher than the conduit only but had no significance in the CI and CN groups (Figure 2M,N). The CNI group showed the largest diameter of regenerated nerves among all the groups in the proximal, medial, and distal sites (Figure 2M,N).

The mean diameters of the medial regions of the CI, CN, and CNI groups were 1.2-fold, 1.4-fold, and 2-fold thicker, respectively, than the C group (Figure 2N). Remarkably, in the medial regions of the regenerated sciatic nerves, the CNI group increased the nerve mean diameter to 1.7-fold and 1.5-fold in comparison with the CI and CN groups, respectively (Figure 2M,N). These results demonstrated that the implantation of human IL12p80 and NSCs within PLA conduits not only improved the functional recovery but also increased the diameter of the regenerated sciatic nerves after the injury.

### 2.3. Human IL12p80 Stimulated Differentiation of Mouse NSCs to Oligodendrocyte Lineages through the Phosphorylation of Stat3

T3/CNTF (a positive control group) and hIL12p80-treated NSCs presented more mature lineages of oligodendrocyte morphology than NSCs controls (KT98/F1B-GFP) (Figure 3A). Quantitative histograms from the Western blotting with specific antibodies recognizing oligodendrocytes and NSCs indicated that the administration of hIL12p80 increased SOX10 and Olig1 protein levels and decreased SOX2 (Figure 3B). The results from Western blotting using the neuronal marker (TuJ1) showed that the βIII-tubulin expression level was not increased in the T3/CNTF or hIL12p80-treated NSCs (data not shown). Moreover, as shown in Figure 3C, quantitative results exhibited that total Stat3 protein level was slightly induced by T3/CNTF or hIL12p80 (100 and 200 ng/mL). The 200 ng/mL dosage of hIL12p80-treated cells significantly increased the Stat3 phosphorylation at Y705 and S727 sites. Taken together, the high concentration of hIL12p80 stimulated the phosphorylation of Stat3 at Y705 and S727 and caused the differentiation of NSCs into oligodendrocyte lineages (Figure 3D).

### 2.4. Implantation of PLGA Conduit 2.1 with NSCs and hIL12p80 Promotes the Functional Recovery and Nerve Conduction of Sciatic Nerve Injury in FVB Mice

In this study, the different constructions of PLGA (C2.0 and C2.1) nerve conduits are shown in Figure 4A. On the eighth week after surgery, the C1.0 (PLA conduit), C2.0, C2.0NI, C2.1, and C2.1NI groups performed significantly better compared to the Neg. group in Rotarod and CMAP analyses (Figure 4B,C). The C2.1NI group had the longest time of Duration on Rotarod and the highest Recovery of CMAP than the C1.0, C2.0, C2.0NI, and C2.1 groups on the eighth week after surgery (Figure 4B,C). In comparison with the C1.0, C2.0, and C2.1 groups on the eighth week after surgery, the C2.0 and C2.1 groups displayed better recovery in the Rotarod and CMAP analyses than the C1.0 group. Implantation of C2.1NI showed a significant difference to the C2.0NI group in the Duration on Rotarod and the Recovery of CMAP of sciatic nerve injury FVB mice (Figure 4B,C). These results demonstrated that implantation of hIL12p80 and NSCs within our novel micro/nanohybrid-structured PLGA conduits without microfibrous structures (C2.1NI group) improved the functional recovery and nerve conduction of sciatic nerve injury FVB mice. Therefore, the PLGA C2.1 nerve conduit was chosen for the subsequent studies.

### 2.5. Implantation of hIL12p80 and NSCs with PLGA Conduit 2.1 Promoted Functional Recovery and Nerve Conduction of Sciatic Nerve Injury in a B6 Mouse Model

Due to the high autotomy (self-mutilation) rate (>60%) in the sciatic nerve injury FVB mouse model, the sciatic nerve injury B6 mouse model was used for evaluation of hIL12p80-induced effects in regenerating sciatic nerve injury. A schematic diagram of the sciatic nerve injury regeneration with PLGA nerve conduits (C2.1) were applied for the implantation surgery (Figure 5A). All of the different treatments for improving the regeneration of injured sciatic nerve are presented in Figure 5B. As shown in Figure 5C, the Neg., C2.1, C2.1I, and C2.1NI groups showed no significant difference in SFI scores on the first two weeks after surgery. On the fourth week after surgery, the C2.1NI group began to show a significantly higher SFI score than the Neg. group (Figure 5C). The C2.1 and C2.1I groups did not show significantly higher SFI scores than the Neg. group until the sixth week after surgery (Figure 5C). In the Rotarod test, the C2.1NI group did not show a significant difference to the Neg., C2.1 or C2.1I groups even at the eighth week after surgery (data not shown). Nevertheless, in the Recovery of CMAP analyses, the C2.1NI group exhibited a significant difference compared with the Neg. and C2.1 groups. The C2.1I group also showed a significant difference comparing with the Neg. group (Figure 5D).

We presented the representative profiles of each group with H&E staining (Figure 6A–C) and measured the mean diameters of regenerated nerves in the proximal, medial, and distal sites of sciatic nerves with the different implantations (Figure 6D). The C2.1NI group showed a significant enhancement of regenerated nerve diameters in proximal, medial, and distal sites compared with the C2.1 group (Figure 6D). The regenerated nerves in the three groups were schematized in Figure 6E. The mean diameters of the medial sites of the C2.1I and C2.1NI groups were twice as large as those of the C2.1 group (Figure 6E). These results demonstrated that implantation of hIL12p80 and NSCs within PLGA C2.1 conduits facilitated the functional recovery through the regeneration of the injured sciatic nerve in B6 mice.

## 3. Discussion

For developing human IL12p80 as a biologic for nerve regeneration, we demonstrated that human IL12p80 and NSCs within PLA or PLGA (C2.0 and C2.1) nerve conduits showed better performance than the C and Neg. groups in the functional assessments and nerve conduction in the sciatic nerve injury FVB and B6 mice. In nerve regeneration evaluation using H&E staining, the diameters of the medial sites of the CNI group were twice those of the C group, and 1.5-fold those of the CN group. When immuno-stained with neuron and oligodendrocyte markers, not only were the diameters of the regenerated sciatic nerves increased, but the oligodendrocytes were also shown to align with the neurons in the proximal-distal axis after the implantation of the conduits with human IL12p80 and mouse NSCs. Our results demonstrated that administration of human IL12p80 can enhance the NSC-improved sciatic nerve regeneration and functional recovery. Therefore, we developed a potential strategy to repair the sciatic nerve injury for clinical application.

Judging from the optical intensities of NF200 and PZ0 staining (Figure 2E–H), it appears that the increased thickness of the nerve was due to an increase in the number of nerve fibers but not the swelling of individual nerve fibers. Our previous results [23] using FGF1 and a rat model showed that when conduits were filled with FGF1, the number of myelinated axons increased from 1704 ± 14 to 3909 ± 136. We showed that in the regenerated nerve (Figure 2E–H), there are numerous NF200-positive cells, suggesting the differentiation of KT98P NSCs into neurons. Schwann cells are the oligodendrocytes in the peripheral nervous system. We have shown, in Figure 2E–L, that NSCs differentiated into Schwann cells in the sciatic nerve via demonstrating the positive staining of PZ0 (the myelinating Schwann cell marker) in the regenerative nerve. For the four treatment groups of C, CI, CN, and CNI, the optical densities for NF-200 staining, normalized to the Conduit-only group, are 1.00, 1.21, 2.54, and 2.21, respectively. The optical densities for PZ0-staining are 1.00, 1.42, 2.01, and 2.09, respectively, while those for DAPI-staining are 1.00, 1.44, 1.50, and 1.72, respectively. The results suggested that exogenously added NSCs are more likely to differentiate into NF-200-positive neurons than PZ0-positive oligodendrocytes (2.54 vs. 2.01). However, when both NSCs and IL12 are included in the treatment (CNI group), the propensity to differentiate into neurons is reduced by 21% from 2.54 to 2.01, while the differentiation toward oligodendrocytes is slightly increased from 2.01 to 2.09. The increased NF200-positive neurons and PZ0-positive oligodendrocytes shown in Figure 2E–L are likely to be the progenies from the transplanted KT98P NSCs. We could not exclude the possibility that some of the progenies were derived from the resident neural progenitor cells in situ. The F1B-GFP NSCs, after differentiation, will cease to express GFP [24]. Therefore, it is not possible to carry out dual-color fluorescence of F1B-GFP and either glial or neuronal markers. In the future, we will employ the F1B-Cre NSCs, which we have established in the lab, to address this question. We are now testing the differentiation capacity of IL12 in human CD133HP NSCs, which our lab previously established [25]. It appears that the oligodendrocyte markers OSP and GalC are both higher in CD133HP when treated with human IL12 (data not shown). We will use the CD133HP human NSCs for future regeneration studies.

Moreover, the human IL12p80-stimulated oligodendrocytes showed an increase in SOX10 and Olig1 protein levels and a decrease in SOX2 protein levels, which were similar to the T3/CNTF-stimulated oligodendrocyte differentiation of NSCs. T3 and CNTF treatments were used as a positive control group in the oligodendrocyte differentiation of NSCs [25,26]. SOX10 is mainly expressed by the oligodendrocyte precursor cells and oligodendrocytes, and regulates the maturation of oligodendrocytes [27,28,29]. SOX10 and Olig1 interact synergistically to induce myelin basic protein expression [30]. SOX10 directs NSCs toward the oligodendrocyte lineage by decreasing the Suppressor of Fused expression [31]. Furthermore, hIL12p80 triggers IL12Rβ1-induced signaling. Stat3 is a member of a family of functionally related Stat proteins that can be activated by various cytokines or growth factors and plays a key role in a variety of biological activities, including cell growth, differentiation, apoptosis, transformation, inflammation, and immune responses [32]. In the canonical IL12 (IL12p70)-induced signaling pathway, IL12Rβ1 provides the high-affinity binding site for IL12p40, which then facilitates the binding of IL12p35 to IL12Rβ2 [33]. The function of IL12p35/IL12Rβ2-induced Stat4 signaling is a key pathway in regulating immune responses [34]. Our results showed that both hIL12p80 and T3/CNTF increased total Stat3 protein levels, and the hIL12p80 stimulated a significant increase in the pStat3-Y705 and pStat3-S727. Human IL12p80 significantly increases pStat3-Y705 five minutes after stimulation (data not shown). We previously showed that WP1066, a Stat3 inhibitor, could reduce mouse IL12p80-induced Stat3 phosphorylation and PZO protein levels [20]. Taken together, we propose a signaling pathway of the hIL12p80-induced oligodendrocyte lineage differentiation of NSCs: IL12p40 subunit of IL12p80 binds to IL12Rβ1 and then induces pStat3-Y705 and dimerization. The dimerized pStat3 then shuttles into the nucleus, turns on gene expression, and promotes oligodendrocyte differentiation. Therefore, administration of human IL12p80 could promote oligodendrocyte differentiation of mouse NSCs, and, in turn, enhance the sciatic nerve regeneration and functional recovery of sciatic nerve injury. Previous reports, including our own, showed that neurotrophic factors (FGF1 [23,35], GDNF [36], and BDNF [36]) that enhanced neuronal differentiation could improve axonal regeneration. This report showed that human IL12, through the enhancement of oligodendrocyte differentiation, could improve axonal regeneration.

In the more advanced constructs of PLGA conduits (including C2.0 and C2.1), the C2.1 and C2.1NI groups showed better recovery in Rotarod and CMAP analyses than the C2.0 and C2.0NI groups, respectively (Figure 4B,C). Although the C2.1NI group showed better functional recovery than the C2.1 group, the opposite results were found in the comparison of C2.0NI and C2.0 groups (Figure 4B,C). Our scanning electron micrographs showed that cells on the flat microfibrous membrane grew along a certain ridge without crossing over to a neighboring ridge [22]. The frozen section images of the NSCs cultured on the C2.0 for three days demonstrated that most cells were grown on the micro-/nanohybrid-structured membrane, and few cells were grown in the inner space and microfiber of the conduit [22]. Although a few cells can grow on the microfiber of the C2.0, the irregular patterning of the microfiber weakened its performance (C2.0 vs. C2.0NI). Since the PLGA C2.1 conduit performed better than PLGA C2.0 and PLA conduits, the C2.1 conduit was therefore chosen for future studies. We were able to show that, on the fourth week after surgery, the C2.1NI group already began to perform better than the Neg. group in SFI scores, but the better performance of the C2.1 and C2.1I groups was delayed until the sixth week (Figure 5C). Thus, the C2.1NI group exerted a speedier recovery than the C2.1 and C2.1I groups. Further, the C2.1NI group outperformed the C2.1 group in the recovery of CMAP analyses (Figure 5D).

During the study, we noticed that the autotomy rate of FVB mice after sciatic nerve injury was high. Rubinstein et al. have demonstrated that B6 mice were highly resistant to autotomy, which suggested that the B6 mice might be a better animal model for peripheral-nerve and whole-limb transplant studies [37]. Here, the autotomy rates in the FVB and B6 mice after sciatic nerve neurotomy were 61.5% and 4.3%, respectively. Therefore, the B6 mouse sciatic nerve model was recommended for future studies. It cannot be ruled out that strain differences in the autotomy rate could be due to an enhanced stress response and, thus, increased glucocorticoid levels. Stress-related changes in glucocorticoid levels have been shown to increase axon regeneration [38]. Therefore, possible differences in glucocorticoid levels in mice of different strains may affect their response to nerve injury.

Even though the autotomy rate for FVB strain is high, those FVB mice that did not undergo autotomy appear to have a fair outcome. Comparing the CNI groups for FVB and B6 strains, SFI is −71.9% vs. −82.8% and CMAP is 58.1% vs. 43.8%. These behavioral differences are present between the two species in spite of the similar increase in the diameters of the regenerative nerves (2-fold vs. 2-fold).

IL12 was known, until now, to only have a role in the immunological setting. Elucidation of IL12 in neuro-regeneration through the induced differentiation toward oligodendrocytes and myelinating Schwann cells in this study is likely to shed light on its therapeutic application in nerve repair. It was established that the defects in oligodendrocytes result in the failure in protecting neurons in ALS [39,40]. Similar impairments were also implicated in other neurodegenerative diseases, including Parkinson’s disease, Alzheimer’s disease [41], and multiple sclerosis [42]. Our demonstration of the role of IL12 in the differentiation of oligodendrocytes and subsequent axonal myelination is likely to lead us to better appreciate the importance of oligodendrocytes in neuronal homeostasis. Our findings could further shed light on IL12′s application not only in damaged nerves but also in rectifying the oligodendrocytes’ defective protection onto neurons in neurodegenerative diseases, including ALS and multiple sclerosis.

## 4. Materials and Methods

### 4.1. Neural Stem Cells Isolation, Neurosphere Formation, and Cell Culture

Mouse neural stem cells (NSCs, KT98) isolation and neurosphere-forming procedures were performed as previously described in our publication [10]. KT98 cells were derived from brain tissue of F1B-Tag transgenic mice and transfected with F1B-GFP plasmids. The F1B-GFP-positive KT98 cells (KT98/F1B-GFP^+^) were sorted using a FACSAria Cell Sorter (BD Bioscience, Franklin Lakes, NJ, USA) and cultured in DMEM/F12 medium containing 10% FBS, 1X penicillin/streptomycin and 500 µg/mL G418 (Merck, Darmstadt, Germany). For neurosphere formation, 1 × 10^5^ KT98/F1B-GFP^+^ cells were seeded in 10-cm dish with neurosphere-forming medium: DMEM/F12 supplemented with 1X B27 (Gibco, Thermo Fisher Scientific, Waltham, MA, USA), 1X Antibiotic-Antimycotic (Gibco, Thermo Fisher Scientific, Waltham, MA, USA), 20 ng/mL EGF (PeproTech, Rocky Hill, NJ,, USA), 20 ng/mL FGF2 (PeproTech, Rocky Hill, NJ, USA), 2 g/mL heparin (Sigma-Aldrich, Burlington, MA, USA), and 500 µg /mL G418 for 7 days. KT98/F1B-GFP^+^ neurosphere-derived cells were used in animal experiments and cell differentiation assays. All cells were cultured at 37 °C with 5% CO_2_.

### 4.2. Sciatic Nerve Injury Regeneration Mouse Model

In this study, FVB/NJ and C57BL/6J male mice (6-8 weeks old) were purchased from the National Laboratory Animal Center and maintained in the National Health Research Institutes Animal Center. All animal protocols followed ethical guidelines and were approved by the Institutional Animal Care and Use Committee (IACUC) of the National Health Research Institutes (Protocol No. NHRI-IACUC-105154A). Animal surgery procedures were in accordance with previous publications [7,20]. In brief, mice were anesthetized by 5% isoflurane (Halocarbon Life Science, North Augusta, SC, USA) inhalation before surgery and 2% isoflurane inhalation during surgery. In sciatic nerve injury surgery, 3 mm of the sciatic nerve segment in the left hindlimb was excised with microscissors, and the homemade PLA [7] or PLGA [22] conduits (length, 5.0 mm; internal diameter, 1.0 ± 0.1 mm) were implanted into the injured nerve site with 1-mm proximal/distal residual nerve stumps inside of conduit ends to connect the 3-mm nerve gap.

The conduit implantation surgical groups included the Conduit only (C) group, Conduit + hIL12p80 (CI) group, Conduit + NSCs (CN) group, and the Conduit + NSCs + hIL12p80 (CNI) group. In the C group, each implanted conduit was filled with a 5 µL mixture of Matrigel (BD Bioscience)/PBS (1:1). In the CI group, the conduit was filled with 5 µL Matrigel/PBS and 100 ng human IL12p80 protein (ab63226, Abcam, Cambridge, UK); in the CN group, conduit was filled with 5 µL mixture with Matrigel/PBS and 1 × 10^6^ NSCs; in the CNI group, conduit was filled with 100 ng of human IL12p80 and 1 × 10^6^ NSCs. In the Sham control (Sham) group, mice underwent the surgery without the excision of sciatic nerves and conduit implantation. In the Negative group (Neg.), mice were excised of a 3 mm sciatic nerve segment in the left hindlimb without conduit implantation.

### 4.3. Functional Assessments: Walking Track Analyses and Rotarod Test

Sciatic functional recovery was assessed using non-invasive methods, including walking track analysis and Rotarod test. Walking track analysis was performed using the Treadmill/TreadScan system (CleverSys Inc, Reston, VA, USA) and presented as the sciatic functional index (SFI) at 1, 2, 4, 6, and 8 weeks after surgery [20]. Adult FVB or B6 mice were used to obtain the normal walking video (total 1500 frames were collected in a complete walking period of one mouse) for TreadScan software calibration (10–12 outlines of each step were sufficient to train the software for identification of the paw position). The SFI is on a scale from −100 to 0, where −100 refers to a complete loss of function and 0 corresponds to the normal walking function.

Rotarod test was executed and thus analyzed using an RT series Rotarod Treadmill (SINGA, Taoyuan, Taiwan) on the fourth and eighth weeks after surgery. Before data collection, mice were trained three times in the program: 0–10 rpm for 90 s, 10–12 rpm for 90 s, and 12–15 rpm for 90 s. For data collection, FVB mice ran on the rotating rod six times at 0–20 rpm for 30 s and 20 rpm for 90 s; B6 mice ran on the rotating rod at 0–36 rpm for 54 s and 36 rpm for 66 s.

### 4.4. Compound Muscle Action Potential Measurement

In this study, the compound muscle action potential (CMAP) was recorded and analyzed with the BIOPAC MP36 and BIOPAC BSL 4.0 software (Biopac Systems Inc., Goleta, CA, USA). The measurement procedures were followed according to our previous publication [20]. Mice were anesthetized with 2% isoflurane inhalation during the CMAP measurement. The stimulating electrodes were placed in the sciatic notch and the recording electrodes were placed in the gastrocnemius muscle (approximately 2 cm from the sciatic notch). Stimulation voltage is 6 volts, 0.1 millisecond, and acquisition length is 200 milliseconds. The Recovery of CMAP was calculated as that for the injured leg divided by that for the control leg.

### 4.5. Hematoxylin and Eosin (H&E) Staining and Immunohistochemistry Staining

On the eighth week after surgery, mice were anesthetized by isoflurane and perfused intracardially with PBS using a peristaltic pump (MP-1000, EYELA, Tokyo, Japan), and then the implanted conduits were collected and fixed with 4% PFA at 4 °C overnight. Fixed samples were equilibrated using 10% sucrose/1X PBS at 4 °C overnight, followed by 30% sucrose/1X PBS at 4 °C overnight, and embedded with Tissue-Tek O.C.T. (Sakura Finetek, Torrance, CA, USA), then frozen in liquid nitrogen and stored in −80 °C. The cryo-embedded nerve conduits were sliced at 10 µm onto slides using a cryostat microtome (MICROM HM550, Thermo Scientific, Kalamazoo, MI, USA) before staining.

For H&E staining, conduit sections on slides were washed with PBS to remove O.C.T compound and stained hematoxylin (HHS16, Sigma-Aldrich, Burlington, MA, USA), washed with tape water, and stained with Eosin Y (Sigma-Aldrich, Burlington, MA, USA) as a counterstain. Samples were dehydrated by 70%, 95%, and 100% ethanol for 3 min, then immersed in xylene (Leica Biosystems, IL, USA) for 5 min and mounted with Sub-X reagent (Leica Biosystems, Richmond, IL, USA). For immunohistochemistry staining, conduit sections were stained with anti-NF200 (1:500, Ab7795, Abcam, Cambridge, UK), and anti-PZ0 (1:500, GTX85466, GeneTex, Irvine, CA, USA) antibodies at 4 °C overnight. After PBS washing, the tissue sections were incubated with Alexa488-conjugated and Alexa594-conjugated secondary antibodies (1:500, Invitrogen, Carlsbad, CA, USA) at room temperature (RT) for 1 h. The tissue sections were stained for nuclei with DAPI (1:5000, Invitrogen) at RT for 3 min and mounted with FluorSave reagent (Merck Millipore, Burlington, MA, USA). All images of regenerated tissue were observed and collected using fluorescent microscopy (Olympus, Tokyo, Japan) and slider scanner (3DH-Pannoramic MIDI, Budapest, Hungary). The diameters of the regenerated nerve were determined by the H&E staining of sciatic nerve sections. “P” and “D” represent the proximal and distal sites of the residual sciatic nerves. “P” is 1.0 mm from the proximal end of the conduit, “D” is 3.0 mm from “P”, and the medial site is 1.5 mm from “P” site.

### 4.6. Cell Differentiation Assay

For NSC differentiation assay, the differentiation condition was modified from our previous publication [20]. Single cells were dissociated from the neurospheres by using 1X HyQTase (Hyclone, GE, Cytiva, IL, USA), and 5 × 10^5^ single cells were seeded into a Poly-D-Lysin (PDL, BD Bioscience)-coated 10 cm tissue culture dish (Corning). The differentiation medium was DMEM/F12 (1:1, Gibco; as a control group) supplemented with or without inducing factors, including thyroid hormone (T3, PeproTech), ciliary neurotrophic factor (CNTF, PeproTech), and hIL12p80 (Abcam). In the T3 + CNTF group, differentiation medium was supplemented with 10 ng/mL T3 and 50 ng/mL CNTF. In the hIL12p80 group, the differentiation medium was supplemented with 100 or 200 ng/mL hIL12p80. The duration of differentiation was thirteen days, and medium was replaced every three days. The differentiated cells were confirmed by morphological observation and Western blot analyses.

### 4.7. Western Blotting

In this study, protein extracts from cell lysates were harvested by 1X RIPA Lysis Buffer (Merck Millipore) supplemented with 1X protease inhibitor cocktail (Roche, Basel, Switzerland). Protein samples (30 μg/lane) were size-fractionated by sodium dodecyl sulfate-polyacrylamide gel (SDS-PAGE) electrophoresis and transferred to polyvinylidene fluoride (PVDF) membranes (Cytiva Amersham, Little Chalfont, UK). Membranes were blocked in blocking buffer with 5% bovine serum albumin (BSA, Bio Basic, Ontario, CA, USA) for 1 h at RT, and incubated with primary antibodies: SOX10 (1:1000, AB212843, Abcam, Cambridge, UK), SOX2 (1:1000, AB5603, Abcam), Olig1 (1:4000, AB124908, Abcam), and β-actin (1:4000, MAB1501, Merck Millipore) in blocking buffer at 4 °C overnight. After PBS washing, membranes incubated with corresponding HRP-conjugated secondary antibodies (1:10,000, Merck Millipore) for 1 h at RT. The signals were detected with ECL reagents (Merck Millipore) and X-ray films. The quantitated data of protein expression levels were analyzed by using the ImageJ software and normalized to the Control group.

To confirm Stat3 phosphorylation, antibodies against Stat3 (1:1000, 9139 S, Cell Signaling), pStat3-Y705 (1:1000, 9131 S, Cell Signaling), pStat3-S727 (1:1000, 9134 S, Cell Signaling,), and β-actin (1:4000, MAB1501, Merck Millipore) were used to determine Stat3 phosphorylation statuses in mouse NSCs after the treatment of hIL12p80.

### 4.8. Statistics

All results were expressed as means ± standard error of the means (SEM). Student’s *t*-test was used for comparing two different treatment groups. For multiple groups comparing, the statistical significance was determined by the one-way analysis of variance (ANOVA) followed by the Bonferroni multiple comparison test by a Statistical Package for the Social Science 13.0 software (SPSS, Chicago, IL, USA). The differences were considered statistically significant when *p* < 0.05. Statistical differences are indicated by *, # *p <* 0.05; **, ## *p* < 0.01; and ***, ### *p* < 0.001.

## 5. Conclusions

In this study, we first demonstrated that the implantation of human IL12p80 and NSCs with PLA or PLGA nerve conduits could improve motor function recovery, promote nerve regeneration, and increase the diameters of newly regenerated nerves in the sciatic nerve injury FVB and B6 mouse models. Considering the self-mutilation rate, the B6 mouse strain is more suitable than FVB as an animal model in the sciatic nerve injury repair experiments. Moreover, PLGA conduits C2.1 are more efficient than PLGA C2.0 and PLA conduits in the functional recovery after sciatic nerve injury. We found that human IL12p80 stimulated mouse NSCs in vitro to differentiate into oligodendrocyte lineages that may be associated with Y705 and S727 phosphorylation of Stat3. Our results could pave the way for developing human IL12p80 as a biologic in repairing sciatic nerve injury, and perhaps ALS and multiple sclerosis in humans.

## Figures and Tables

**Figure 1 ijms-23-07002-f001:**
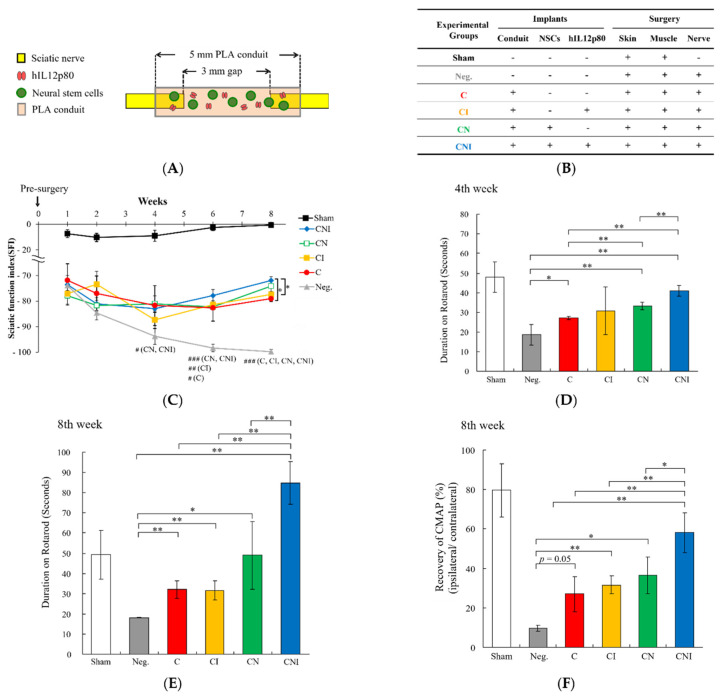
Implantation of a conduit with NSCs and human IL12p80 improves functional recovery and nerve conduction in sciatic nerve injury mice. (**A**) Schematic diagram and illustration of implanted PLA conduit, filled with matrix gel, NSCs, and hIL12p80. (**B**) Description of different experimental groups treated with or without the implants of conduit, NSCs, and hIL12p80 and with or without the surgery of skin, muscle, and nerve of a mouse model. (**C**) Sciatic functional index (SFI) was calculated to assess the functional recovery of sciatic nerves. The hashtag (#) indicated that, beginning at the fourth week, the four different treatment groups showed a progressively significant difference when compared with the Neg. group. The asterisk (*) indicated that, in the eighth week, the CNI group showed a statistical difference with the C and CI groups. (**D**) The quantitative histograms of the Duration on Rotarod on the fourth week. The duration time on the Rotarod of mice of the CNI group was shown to be significantly longer than the Neg., C, and CN groups, but not the Sham and CI groups. (**E**) The quantitative profile of the Duration on Rotarod on eighth week after surgery. Only the Neg. group showed significantly less time on Rotarod than the Sham group. The CNI group showed longest time on Rotarod in comparison with the other groups. (**F**) The CMAPs of the injured leg and contralateral leg were measured. The Recovery of CMAP was calculated as injured leg divided by control leg. The CNI group showed the highest of the Recovery of CMAP in comparison with the Neg., C, CI, and CNI groups. These results are shown as mean ± SEM (n = 3 in Neg., C, and CI groups, n = 4 in Sham and CN groups, and n = 5 in CNI group). Statistical differences are indicated by *, # *p* < 0.05, **, ## *p* < 0.01, and ### *p* < 0.001.

**Figure 2 ijms-23-07002-f002:**
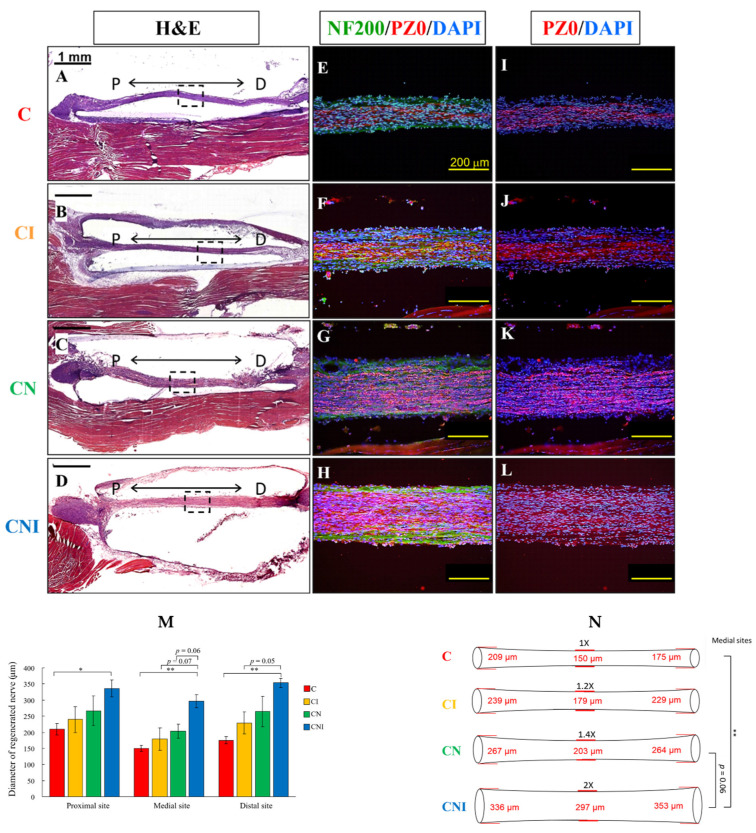
Implantation of PLA conduits with NSCs and human IL12p80 promoted nerve regeneration in the sciatic nerve injury FVB mouse model. (**A**–**D**) Hematoxylin and eosin (H&E) staining of tissue sections was carried out for the measurements of diameters in sciatic nerve regeneration. “P” and “D” represent the proximal and distal sites of the residual sciatic nerves (“P” is 1.0 mm from the proximal end of the conduit and “D” is 3.0 mm from “P”). (**E**–**L**) Immunohistochemistry staining with antibody of NF200 (nerve fibers, green), PZ0 (myelin Schwann cells, red), and DAPI (nucleus, blue). For highlighting the myelin sheath cells without NF200 staining, each panel I-L corresponded to E–H, respectively. (**M**) The quantitative results of regenerated nerve diameters at proximal, medial, and distal sites. The medial sites were defined as the middle between the proximal and distal sites (n = 3 in the C and CI groups, n = 4 in the CN group, and n = 5 in the CNI group). (**N**) The illustration of the regenerated sciatic nerves with the different treatment groups. Scale bars in (**A**–**D**) are 1 mm and (**E**–**L**) are 200 μm. Statistical differences are indicated by * *p* < 0.05 and ** *p* < 0.01.

**Figure 3 ijms-23-07002-f003:**
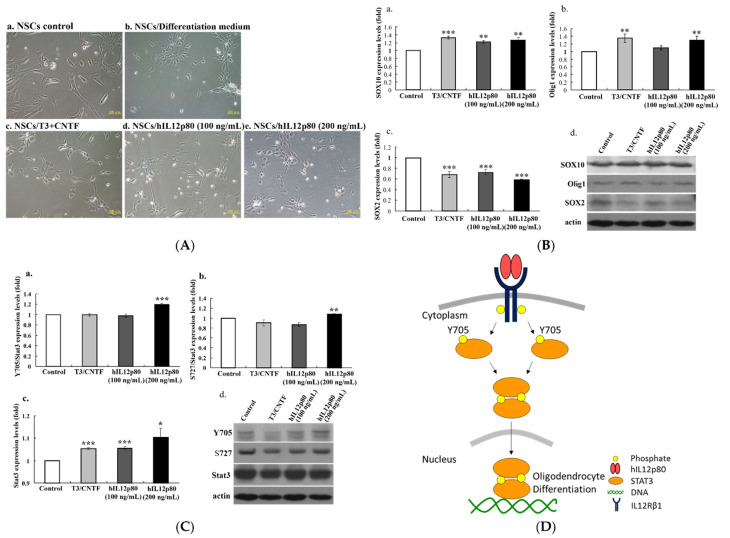
Human IL12p80 stimulated differentiation of mouse NSCs to oligodendrocyte lineages through phosphorylation of Stat3 in vitro. (**A**) The morphology of the differentiated cells from mouse NSCs to oligodendrocytes is due to the addition of T3/CNTF or hIL12p80. (**B**) Quantitative results and a representative profile of immunoblotting with oligodendrocyte markers: (**a**) SOX10, (**b**) Olig1 and a neural stem cell marker: (**c**) SOX2. The Western blot results are shown in Figure (**d**). (**C**) Quantitative results and a representative profile of immunoblotting with the Stat3-Y705 and S727 phosphorylation. (**D**) The illustration of the potential signaling pathway of mouse NSCs differentiated into oligodendrocyte lineages through the hIL12p80-induced phosphorylation of Stat3. Statistical differences compared to the control group are indicated by * *p* < 0.05, ** *p* < 0.01, and *** *p* < 0.001.

**Figure 4 ijms-23-07002-f004:**
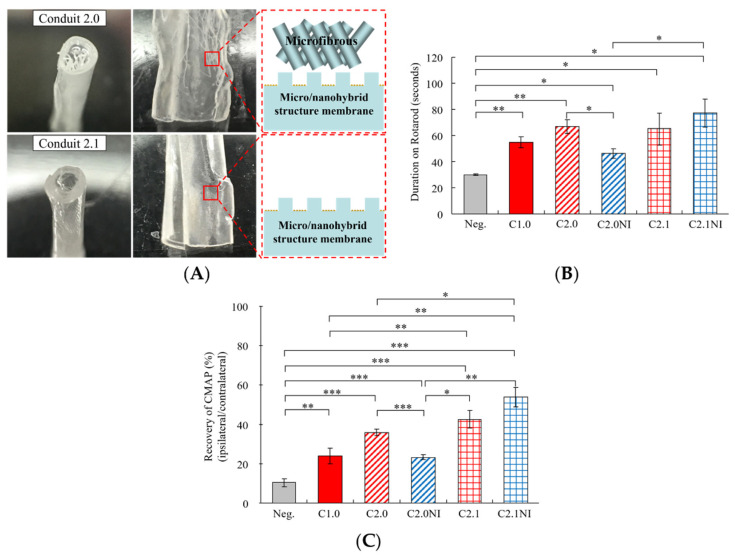
The construction design of PLGA conduits and functional assessments by using sciatic nerve injury FVB mouse model. (**A**) The microscopic views and illustration of structures in Conduit 2.0 and Conduit 2.1. (**B**) Quantitation of the Duration on Rotarod in the various treatment groups in the eighth week after surgery. (**C**) Quantitation of the Recovery of CMAP of the different treatment groups. The CMAP values of the injured and contralateral legs were measured in the eighth week after surgery. All data are presented as a mean ± SEM (n = 3 in Neg., C1.0, C2.0, C2.1, and C2.1NI groups, and n = 4 in C2.0NI group). Statistical differences are indicated by * *p* < 0.05, ** *p* < 0.01, and *** *p* < 0.001.

**Figure 5 ijms-23-07002-f005:**
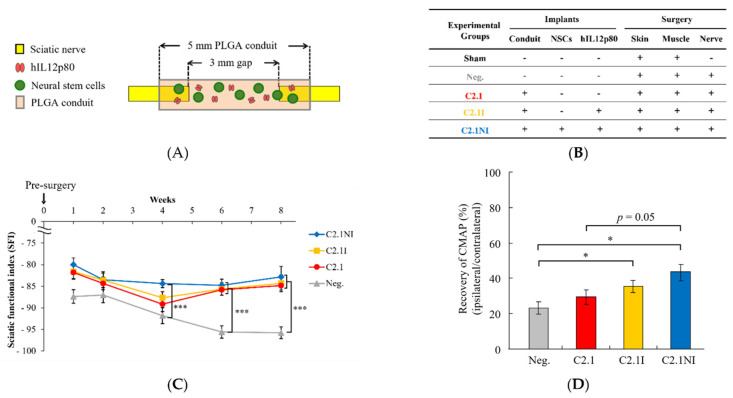
Summary of the various treatments and functional assessments for the implantation of PLGA conduits with human IL12p80 and NSCs in the sciatic nerve injury B6 mouse model. (**A**) The illustration of a micro/nanohybrid-structured PLGA nerve conduit C2.1, which was filled with hIL12p80, matrix gel, and NSCs. (**B**) The summary table of the different treatment groups with or without implantation of Conduit 2.1, NSCs, and hIL12p80, with or without the surgery of skin, muscle, and nerve, and with or without the nerve transection in B6 mouse model. (**C**) SFI was calculated to assess functional recovery. The C2.1, C2.1I, and C2.1NI groups showed a significant difference compared with the Neg. group in statistical analysis in the sixth and eighth weeks. In the fourth week, only the C2.1NI showed a significant difference compared with the Neg. group. (**D**) The percentage of the Recovery of CMAP in the different treatment groups. All data are presented as mean ± SEM (n = 5 in C2.1, C2.1I, and C2.1NI groups, and n = 6 in Neg. group). Statistical differences are indicated by * *p* < 0.05, and *** *p* < 0.001.

**Figure 6 ijms-23-07002-f006:**
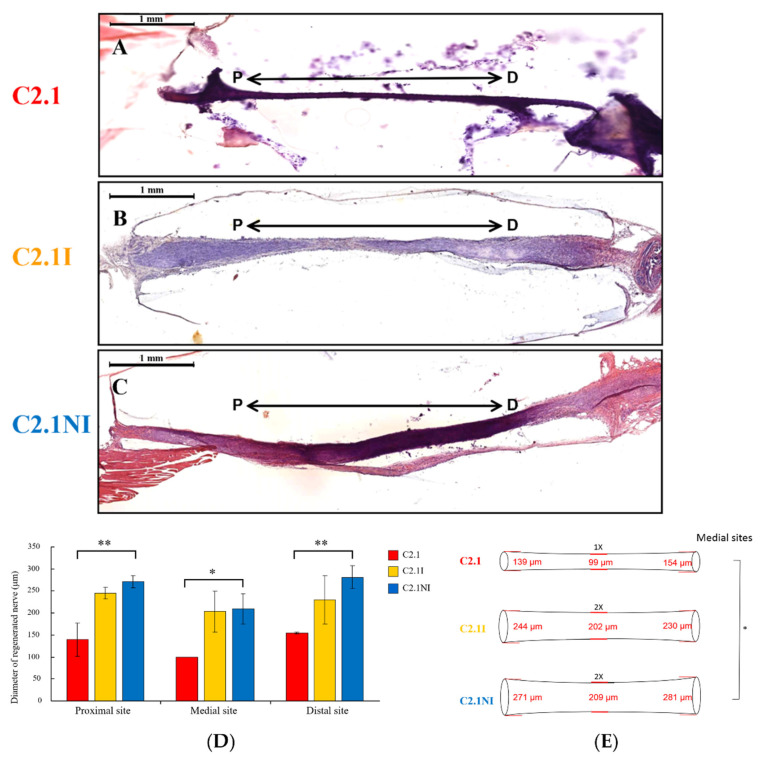
Implantation of PLGA conduits with human IL12p80 and NSCs promoted nerve regeneration in the sciatic nerve injury B6 mouse model. (**A**–**C**) H&E staining of the tissue sections for the measurements of the diameters in sciatic nerve regeneration. “P” is 1.0 mm from the proximal end of the conduit and “D” is 3.0 mm from “P”. (**D**) The quantitative results of regenerated nerve diameters at the proximal, medial, and distal sites. All data are presented as mean ± SEM (n = 2 in C2.1 and C2.1I groups, and n = 3 in C2.1NI group). (**E**) The illustration of the sciatic nerves in the different treatment groups. Scale bars in (**A**–**C**) are 1 mm. Statistical differences are indicated by * *p* < 0.05 and ** *p* < 0.01.

## Data Availability

Data supporting the findings of this study are available within the figures and the Section 2. Further data are available from the corresponding author upon reasonable request.

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
