# Peer review of "Human IL12p80 Promotes Murine Oligodendrocyte Differentiation to Repair Nerve Injury"

_ijms, 2022, doi:10.3390/ijms23137002_

Round 1
Reviewer 1 Report
Major concerns
Authors present a study with a very similar setup as many other studies before (nerve regeneration model in rats, conduit + coating), and the results they can obtain are not considerably better. Many previous studies have shown some early (4 wks) improvement of SFI, with no difference to the controls in the end. Authors should emphasize what makes their results better than those prior results.
Rat nerve regeneration is known to take place very well over small distances (<3mm), relatively independent from conduit material and coating.
And as shown from many similar experiments, SFI is seldom different between experimental groups after eight weeks. Can authors only effect some earlier improvement, but no better regeneration in the end?
Authors show improved nerve regeneration over a 3mm distance by addition of IL-12 and murine oligodendrocytes. Can authors extrapolate on larger nerve gaps, or on results in other species than rodents from the results obtained here?
Minor concern:
For enhancement of peripheral nerve repair, differentiation of NSCs into Schwann cells rather than into oligodendrocytes could be expected to be desirable. Please comment onto that.
Reviewer 2 Report
The article by Chung et al. is aimed at establishing whether human Interleukin 12 (hIL12) favors nerve regeneration in a mouse model of sciatic nerve injury. This work is an extension of previously published experiments (https://doi.org/10.1016/j.mcn.2016.11.007) where mouse IL12, instead of human IL12, had been used. It confirms that the use of Poly(L-lactic-co-glycolic acid) (PLGA) conduits filled with hIL12 plus mouse-derived neural stem cells improves nerve recovery more than either neural stem cells alone or IL12 alone. Although the study lacks substantial novelty, it may be useful in the clinical field of peripheral nerve reconstruction. In this connection, however, there are some issues that the authors should take into account.
Source of neural stem cells. It seems to me that the main purpose of this study was to obtain translational information that my foster the design of clinical protocols for human use. The authors tested here the effect of mouse neural stem cells added to the conduit in the presence of hIL12. Does hIL12 favor acquisition of an oligodendrocytic phenotype in human neural stem cells? If so, I suggest to test in the mouse model the efficacy of human neural stem cells added to the conduit in conjunction with hIL12.
Phenotype acquired by mouse neural stem cells in culture. The study examined the effect of hIL12 on mouse neural stem cell differentiation in vitro and found that treatment increases the protein levels (WB) of oligodendroglial markers, such as Sox10 and Olig1, but not of the neuronal protein TuJ1. It is somewhat surprising that there were no cells differentiated into neurons. How can this be explained? In addition, I suggest using immunocytochemistry in addition to WB because IHC allows quantification of the percentage of cells that differentiate into glia and neurons (and of undifferentiated progenitors).
Destiny of injected mouse neural stem cells. What is the destiny of the neural stem cells added to the conduits after 8 weeks? Do they retain proliferation capacity? Are they differentiated into oligodendrocytes and/or neurons? I suggest using IHC in sections across the implant to quantify FB1-GFP-positive cells that are proliferating (e.g. by using Ki-67 IHC) or have undergone differentiation into either glial or neuronal cells. This information would allow one to establish whether the neural stem cells added to the conduit retain stemness and whether they show propensity to differentiate into oligodendrocytes in the conduit niche.
Quantification of regeneration. Sections in the conduit region were stained for the nerve fiber marker NF-200 and the Schwann cell myelin marker PZ0 in order to evaluate nerve regeneration. Methods do not explain whether the nerve thickness was measured in sections processed for NF-200, PZ0 or both. Measurements of the thickness of the nerve within the conduit indicates a certain improvement at the end of treatment with hIL12 plus neural stem cells (Fig. 2). It remains to be established whether the thickness increase was due to an increase in the number of nerve fibers or to swelling of individual nerve fibers. In order to clarify this issue individual fibers should be counted by using electron microscopy. However, a rough quantification of nerve fiber density could be attempted in the available sections by evaluating the optical density for NF-200. Moreover, these sections may be used for counting the number of cells positive for both DAPI and PZ0 in order to establish the effects of the different experimental condition on the recruitment of Schwann cells.
Mouse strain. In this and their previous study https://doi.org/10.1016/j.mcn.2016.11.007), the authors used as model FVB mice. At a certain point of the article the authors state that these mice are very prone to autotomy (>60%; lines 506-511) and therefore decide to move to another mouse strain (C57BL/6J) that is reported to exhibit a low autotomy rate. Aside the fact that this statement is supported by a wrong citation (citation #9 does not provide this information) it is surprising that data regarding the high autotomy rate of FVB mice had not been reported in their previous publication. This is not a secondary issue because, although strain differences in the autotomy rate remain to be elucidated, an enhanced stress response and, thus, increased glucocorticoid levels in mice prone to autotomy cannot be ruled out. Stress-related changes in glucocorticoid levels have been shown to increase axon regeneration (https://doi.org/10.1523/ENEURO.0246-17.2017). Therefore, possible differences in glucocorticoid levels in mice of different strains may affect their response to nerve injury. This point should be discussed. In the absence of information on blood levels of glucocorticoids in FVB and C57BL mice after nerve injury, the authors could at least statistically compare data obtained in FVB and C57BL mice subjected to the same experimental treatments (type of conduit and addiction of IL12 and/or neural stem cells) in order to establish differences/similarities.
Sex of mice. The sex of mice subjected to the different treatments should be specified because there is evidence for gender differences in nerve regeneration (e.g. https://doi.org/10.1186/1471-2202-15-107; https://doi.org/10.1016/j.expneurol.2004.05.015). Gender differences in regeneration should be mentioned in the text.
Statistics. The simple T-test is not appropriate, considering that there were more than 2 experimental groups. Comparison of a given group with each of the other groups may provide erroneously significant differences. Data should be analyzed with a multiple comparison test, such as Fisher’s test, after ANOVA
Minor
Lettering in Fig. 3 is too small and very difficult to read
Numerous grammar errors in the text make it very difficult to read and understand.
Round 2
Reviewer 1 Report
All of my concerns have been adequately addressed. I recommend acceptance in the present form.
Author Response
Thank you very much!
Reviewer 2 Report
The Authors have improved the manuscript but there are some points that still need to be addressed,
As answer to the comment regarding the “Phenotype acquired by mouse neural stem cells in culture” the authors state that “We did not test if the anti-TuJ1 positive cells were present in the Western blots”. This does not make much sense because WB can simply show the expression of proteins and not the presence of cells. As already suggested, results would be more convincing if the authors carried out immunohistochemistry for neuronal and glial markers and counted the relative number of cells of each phenotype.
The authors have answered only partially to the comment regarding “Destiny of injected mouse neural stem cells”. I suggested to use IHC for Ki-67 because, in my opinion, it is important to establish whether the cells added to the conduit retain or not proliferation capacity. IHC for Ki-67 could be easily carried out using sections already available because Ki-67 is an endogenous marker of cycling cells.
The authors have answered only partially to the comment regarding “Quantification of regeneration”. I think that with the material already available the authors could easily quantify nerve fiber density by evaluating the optical density for NF-200. In addition, they could easily evaluate the number of cells positive for both DAPI and PZ0. This evaluation is of relevance in order to establish the effects of different experimental conditions on the recruitment of Schwann cells.
The authors have answered only partially to the comment regarding “Mouse strain”. Given the potential relevance of the mouse train why did the authors not compare data obtained in FVB and C57BL mice subjected to the same experimental treatments as suggested? This could be easily done with the data already available.
Statistics. The Authors state that “Student’s t-test was used for comparing two groups and one-way ANOVA was used for comparing multiple groups”. This is not correct. ANOVA test tells you whether you have an overall difference between your groups, but it does not tell you which specific groups differed – post hoc tests do. The t-test is not correct when analyzing multiple groups. Ad already suggested, the authors should use a post-hoc test for multiple comparisons (such as Fisher’s or Tukey) after ANOVA. In addition, they should report in the text the values of the F, the degrees of freedom between and within groups and the p values of ANOVA for each analyzed variable.
